# A Systematic Evaluation of Node Embedding Robustness

**Alexandru Mara**
Ghent University, Ghent, Belgium
`alexandru.mara@ugent.be`

**Jefrey Lijffijt**
Ghent University, Ghent, Belgium
`jefrey.lijffijt@ugent.be`

**Stephan Günnemann**
Technical University of Munich
`guennemann@in.tum.de`

**Tijl De Bie**
Ghent University, Ghent, Belgium
`tijl.debie@ugent.be`

## Abstract

Node embedding methods map network nodes to low dimensional vectors that can be subsequently used in a variety of downstream prediction tasks. The popularity of these methods has grown significantly in recent years, yet, their robustness to perturbations of the input data is still poorly understood. In this paper, we assess the empirical robustness of node embedding models to random and adversarial poisoning attacks. Our systematic evaluation covers representative embedding methods based on Skip-Gram, matrix factorization, and deep neural networks. We compare edge addition, deletion and rewiring attacks computed using network properties as well as node labels. We also investigate the performance of popular node classification attack baselines that assume full knowledge of the node labels. We report qualitative results via embedding visualization and quantitative results in terms of downstream node classification and network reconstruction performances. We find that node classification results are impacted more than network reconstruction ones, that degree-based and label-based attacks are on average the most damaging and that label heterophily can strongly influence attack performance.

## 1 Introduction

In recent years, the design of robust machine learning models has become an important topic and attracted significant amounts of research attention [1–4]. The term 'robust' refers to the ability of a model to provide consistent and accurate predictions under small perturbations of the input data. These perturbations can appear in the form of random noise, out of distribution (OOD) data, or partially observed inputs [5]. They can affect models at train or evaluation times and be random or adversarial in nature. For a more complete overview of robustness in machine learning we refer the reader to [6]. In this manuscript, we empirically study both random and adversarial attack scenarios where perturbations are either a consequence of noise or specifically crafted to reduce model performance. We further focus our analysis on attacks affecting the models at training time exclusively, also know as the poisoning scenario [7].

Simultaneously, node representation learning or node embedding models have become increasingly popular for bridging the gap between traditional machine learning and network structured data [8–10]. These approaches map network nodes to real-valued vectors that can be subsequently used in downstream prediction tasks such as classification [11] and regression [12]. Training of these models can be performed in a semi-supervised or unsupervised fashion. In the former, embeddings are optimized for a particular downstream task while in the latter, general purpose embeddings are obtained. Robustness is an important feature for representation learning models as well. One would generally prefer small changes in the input networks to have a minimal impact on the vector representations learned and on downstream task performance. Moreover, with the deployment of these models in safety-critical environments (e.g., [13]) and on the web (where adversaries are common [14, 15]), robustness evaluation has become ever more essential. Unfortunately, the robustness of

A. Mara et al., A Systematic Evaluation of Node Embedding Robustness. *Proceedings of the First Learning on Graphs Conference (LoG 2022)*, PMLR 198, Virtual Event, December 9–12, 2022.

unsupervised node embedding approaches is poorly understood. Some recent studies have analyzed particular semi-supervised models based on the Graph Neural Network (e.g., [16]) and on shallow models (e.g., [17]). Others, have evaluated specific unsupervised random walk approaches under poison attacks [7]. Methods leveraging unsupervised embedding learning paradigms such as matrix factorization and deep neural networks, have not received much attention yet. Additionally, there is a lack of studies providing broader robustness evaluations and comparing multiple models.

We perform a systematic empirical analysis of the robustness of foundational works in the field of node embeddings. Among the 9 unsupervised approaches evaluated we include Node2vec [18], GraRep [19], and SDNE [20], which have inspired many other methods based on similar principles, e.g., [21–23]. The models considered can be categorized into Skip-Gram, matrix factorization, and deep neural networks, and their robustness is compared on two downstream tasks: node classification and network reconstruction. We evaluate robustness under randomized and adversarial attacks targeting the network edges. For adversarial attacks we limit the scope to heuristic-based approaches where edges are targeted based on topological network properties (e.g. assortativity, degree). In contrast, optimization-based attacks (e.g. [7, 24–26]) solve a multi-level optimization problem to identify the most promising targets. Heuristic attacks are, thus, simpler and more computationally efficient making them more easily accessible to an attacker. Additionally, they do not require tailoring to specific embedding models and downstream tasks –as many optimization-based approaches do– and provide intuitive and explainable targets. Moreover, the heuristic attacks considered in this manuscript have already shown to effectively lead to structural collapse in networks [27]. The analysis of stronger optimization-based models is left for future work. Lastly, we focus our evaluation on global attack scenarios where changes can be made to the entire graph structure provided a fixed attack budget.

**Contributions.** Our main contribution is a systematic analysis of node embedding robustness. We evaluate a total of 9 unsupervised node embedding approaches based on three learning paradigms. We employ 6 small and mid-sized networks and compare 14 different poison attack strategies. Further, we investigate differences between randomized and adversarial attacks and compare edge addition, deletion and rewiring strategies. We also investigate attacks leveraging full knowledge of the node labels, commonly used as baselines, and show that network homophily (tendency of nodes with similar labels to be connected) and heterophily (where nodes of different labels are more often connected) have a strong impact on their performance. This work constitutes the first empirical evaluation of its magnitude on node embedding robustness.

The remainder is organized as follows: in Section 2 we present the related work and in Section 3 we introduce the embedding methods and attack strategies evaluated. In Section 4, we discuss the experimental evaluation and results and finally, in Section 5 we outline our main conclusions.

## 2 Related Work

A large body of research has shown that traditional machine learning models and more recently deep neural models can be easily misled into providing wrong answers with high confidence [28, 29]. Work on identifying and protecting against these adversarial attacks has particularly developed in the field of computer vision [30]. Works in this field, including [31, 32], have also shown how changes unperceivable to the human eye can result in dramatic performance drops or misclassifications. Later, adversarial attacks were introduced in the field of network science [5]. In [27], the authors show how structural properties of networks can collapse as a result of attacks. The authors further provide a framework for simulating attacks and defenses on networks. With the popularization of node embedding methods authors have also investigated adversarial attacks on particular semi-supervised [16, 33] and unsupervised [17] approaches. A survey reviewing various adversarial attacks and defense strategies, specifically designed for semi-supervised representation learning models, is presented in [34]. The authors also collect implementations of representative attack and defense strategies and make them publicly available as a PyTorch toolbox. For unsupervised representation learning models, while there are some empirical studies comparing performance (e.g., [35]), there is little research evaluating robustness. With the present work, our aim is to fill this gap and provide a fist empirical study and overview on the robustness to random and adversarial attacks of unsupervised node embedding approaches. Lastly, akin to the results in [36] for graph classification using GNN models, we also investigate patterns in the attacks and relations to the network structure. This analysis reveals an interesting effect of label heterophily on node classification attacks.

## 3 Methods

In this section we introduce the node embedding approaches evaluated and the attack strategies used to poison the input networks. Regarding notation, in what follows we will use $\mathbf{G} = (\mathbf{V}, \mathbf{E})$ to refer to an undirected graph with vertex set $\mathbf{V} = \{v_1, \ldots, v_N\}$, $N = |\mathbf{V}|$ and edge set $\mathbf{E} \subseteq (\mathbf{V} \times \mathbf{V})$, $M = |\mathbf{E}|$. We will represent edges or connected node-pairs as unordered pairs $\{v_i, v_j\} \in \mathbf{E}$. And refer to pairs $\{v_i, v_j\} \notin \mathbf{E}$ as non-edges or unconnected node-pairs. Node embeddings are denoted as $\mathbf{X} = (\mathbf{x}_1, \mathbf{x}_2, \ldots, \mathbf{x}_N)$, $\mathbf{X} \in \mathbb{R}^{N \times d}$ where $\mathbf{x}_i$ is the d-dimensional vector representation corresponding to node $v_i$.

### 3.1 Node embedding methods

For our experimental evaluation we have selected 9 representative methods spanning three different embedding learning paradigms, namely Skip-Gram, matrix factorization and deep neural networks. Next, we introduce each paradigm and the corresponding methods.

**Skip-Gram.** These approaches capture node similarities in the graph through random walks and leverage the Skip-Gram model [37] to obtain node representations that maximize the posterior probability of observing neighboring nodes in the walks. From this category we evaluate: Deepwalk [38], the seminal work that proposed fixed length random walks to capture node similarities and Skip-Gram (approximated via hierarchical softmax) for learning the embedding matrix $\mathbf{X}$; Node2vec [18], which introduced more flexible random walks controlled by in/out and return parameters and approximates Skip-Gram via negative sampling; LINE [39], where the authors leverage first and second order proximities to learn representations; And finally, VERSE [11], which minimizes the KL-divergence between a similarity metric on $\mathbf{G}$ (by default Personalized PageRank) and a vector similarity on $\mathbf{X}$.

**Matrix Factorization.** Factorization methods take as input node similarities encoded in the graph Laplacian, incidence matrices, adjacency matrices ($\mathbf{A}$) and their polynomials, etc. and compute low dimensional embeddings by factorizing the selected matrix. We evaluate the following methods based on this paradigm: GraRep [19], HOPE [40], NetMF [41] and M-NMF [42]. GraRep factorizes high order polynomials of $\mathbf{A}$, HOPE can factorize different similarity matrices provided they can be expressed as a composition of two sparse proximity matrices. NetMF decomposes the Deep-Walk transition matrix via SVD and lastly, M-NMF computes embeddings via non-negative matrix factorization and incorporates community structure in this process.

**Deep Neural Networks.** Deep neural models, from auto-encoders to Siamese networks or CNNs, have also been used to obtain node representations from a graph's link structure in an unsupervised fashion. Among these types of methods we evaluate SDNE [20], a deep neural model that captures first and second order proximity in the graph.[1]

### 3.2 Network attacks

We subdivide network attacks into randomized and adversarial and further into three main types based on the changes to the network structure. These changes are edge addition, edge deletion and edge rewiring. Table 1 summarizes all attacks and below we briefly describe each one.

**Randomized Attacks.** These attacks are designed to simulate random errors or failures in the networks. We consider edge addition (*add_rand*), deletion (*del_rand*) and rewiring (*rew_rand*). In the first case, pairs of nodes, $v_i, v_j \in \mathbf{V}$ are selected uniformly at random and added to $\mathbf{E}$ iff $v_i \neq v_j$ and $\{v_i, v_j\} \notin \mathbf{E}$. For deletion attacks, edges $\{v_i, v_j\} \in \mathbf{E}$ are selected uniformly at random and removed from $\mathbf{E}$ iff $d_i \geq 2 \wedge d_j \geq 2$. Here $d_i$ and $d_j$ represent the degrees of nodes $v_i$ and $v_j$, respectively. In rewire attacks we use *del_rand* to remove a budget of edges $\{v_i, v_j\} \in \mathbf{E}$ and then reconnect each $v_i$ to a new node $v_k$ such that $v_k \neq v_j$ and $\{v_i, v_k\} \notin \mathbf{E}$.

**Adversarial Attacks.** We also consider a particular type of heuristic-based adversarial attacks which target specific network properties such as node degrees, assortativity, and node labels. Despite their lower effectiveness compared to optimization-based attacks, we evaluate these approaches

---

[1]We also evaluated PRUNE [12] but despite our best efforts the method severely underperformed on all tasks.

**Table 1:** Poison attacks evaluated and their types: (D) deterministic, (ND) non-deterministic.

| Edge addition | | Edge deletion | | Edge rewiring | |
|---|---|---|---|---|---|
| Name | Type | Name | Type | Name | Type |
| add_rand | ND | del_rand | ND | rew_rand | ND |
| add_deg | ND | del_deg | D | - | - |
| add_pa | ND | del_pa | D | - | - |
| add_da | ND | del_da | D | - | - |
| add_dd | ND | del_dd | D | - | - |
| add_ce | ND | del_di | ND | DICE | ND |

due to their lower computational complexity, applicability to different embedding methods and downstream tasks, and explainable attack targets. Moreover, they aim to modify key structural properties commonly captured by representation models and can thus lead to worse representations. The heuristics considered have also been successfully used as baselines in previous works e.g., [7, 27].

For all edge addition attacks we ensure that newly generated pairs do not already exist in the graph, i.e., $\{v_i, v_j\} \notin \mathbf{E}$, and they do not form selfloops, i.e., $v_i \neq v_j$. For degree-based (*add_deg*) and preferential attachment (*add_pa*) edge addition strategies we sample nodes uniformly and based on degree, respectively, and connect them to destination nodes sampled based on degree. For the degree assortativity (*add_da*) and disassortativity (*add_dd*) attacks, we generate edges which increase/decrease this property. We define assortativity ($r$) akin to [43] and compute it per edge as the product of standard scores of $d_i$ and $d_j$, i.e. $r_{\{v_i, v_j\}} = (d_i - \mu)/\sigma \cdot (d_j - \mu)/\sigma$. Where $\mu = \frac{1}{M} \sum_{l=1}^{M} d_l^2$ and $\sigma = (\frac{1}{M} \sum_{i=1}^{M} d_i \cdot (d_i - \mu)^2)^{1/2}$. Thus, to increase assortativity we sample nodes $v_i$ with probability $p_i \propto |d_i - \mu|$ and connect them to nodes $v_j$ sampled with probability $p_j \propto \frac{1}{|d_i - d_j|}$. To increase disassortativity we sample nodes $v_i$ as above and $v_j$ with $p_j \propto |d_i - d_j|$. The *add_ce* strategy applies to attributed graphs only and adds a set of random edges connecting nodes of dissimilar labels, exclusively.

Unless otherwise specified, edge deletion attacks ensure that input networks do not become disconnected after the attack. For *del_deg* and *del_pa* we first sort all edges based on the appropriate metric, i.e., $d_i + d_j$ for *del_deg* and $d_i \times d_j$ for *del_pa*, and later remove the top edges that do not disconnect the network. For *del_da* and *del_dd* we compute $r_{\{v_i, v_j\}}$ and $-r_{\{v_i, v_j\}}$ as described above. Then, we sort the edges based on these properties and take the top candidates in each case while avoiding disconnections. The *del_di* strategy applies exclusively to attributed graphs and randomly selects edges for removal where the incident nodes share the same label.

Finally, *DICE* [7] is an adversarial attack where edges are removed or added to a network with equal probability. Edges are removed according to the *del_di* strategy and added following *add_ce*. It is important to note that all edge deletion attacks with the exception of *del_di* are deterministic while the remaining addition and rewire attacks are non-deterministic (see Table 1).

## 4   Experiments

In this section we present the experimental setup, networks used and the results obtained. All our experiments were carried out on a single machine equipped with two 12 Core Intel(R) Xeon(R) Gold processors, 1TB of RAM and an RTX 3090 GPU.

To ensure reproducibility of results, we have employed and extended the capabilities of the EvalNE toolbox [44]. This Python framework allows users to assess the performance and robustness of network embedding approaches for downstream node classification, network reconstruction, link prediction and sign prediction. In the framework we have integrated a variety of random and adversarial poison attack strategies, including those introduced in Section 3.2 and Table 1. In EvalNE, complete evaluation pipelines and hyperparameters are specified through configuration files which can be used at any time to replicate results. These configuration files together with the rest of our code are available online at `https://github.com/aida-ugent/EvalNE-robustness`.

## 4.1 Preliminaries and Setup

As pointed out in Section 1, the main goal of this paper is to investigate the robustness of node embedding approaches to poison attacks. To this end we report changes in downstream node classification and network reconstruction performances for different attacks on the input graphs. Next, we summarize the main goals and evaluation pipelines for both tasks and the overall evaluation setup.

**Node Classification.**    Given an input graph and labels for a subset of vertices, the goal in node classification is to infer the labels of the remaining vertices. To evaluate node classification robustness we proceed as follows. (1) We start by attacking an input network $\mathbf{G}$ with a specific strategy (from Table 1) and budget $b$. The budget defines the number of edges an attacker can add, delete or rewire in the network, expressed as a fraction of the total edges. For example, $b = 0.1$ indicates 10% of all edges in $\mathbf{E}$. (2) The attacked network $\hat{\mathbf{G}} = (\mathbf{V}, \hat{\mathbf{E}})$ is then provided as input to a node embedding approach which yields a representation matrix $\mathbf{X}$ containing vertex representations as its rows. As shown by Mara et. al. [35], gains from optimizing the hyperparameters of these models are marginal, and thus, we resort to fixed default values.[2] We also fix the embedding dimensionality $d = 128$. (3) Given a number of training nodes $N_{tr}$ (also defined as a fraction of all nodes in $\mathbf{V}$), a multi-class one-versus-rest Logistic Regression classifier with 5-fold cross validation is trained to predict node labels from node representations. (4) We repeat the previous step 3 times with different node samples and report average results. For some experiments we will report results independent of the value of $N_{tr}$. In these cases we additionally average results over several values of $N_{tr}$. (5). Finally, and unless otherwise specified, for the non-deterministic attacks listed in Table 1 we repeat the complete process 3 times with varying random seeds resulting in different sets of edges being removed in step 1). We report node classification performance in terms of f1_micro and f1_macro.

**Network Reconstruction.**    In this task the aim is to investigate how well the link structure of an input network can be recovered from the node representations. To this end node representations are first learned from the input network. Then, node-pair representations are derived by applying a binary operator on the node representations. Finally, a binary classifier is trained to discriminate edges from non-edges. High quality representations are expected to result in the classifier scores of edges being higher than those of non-edges.

We evaluate robustness on this task akin to node classification. (1) We attack the input network $\mathbf{G}$ with a given strategy and budget $b$. (2) We compute node representations for $\hat{\mathbf{G}}$ with different methods for which we use fixed default hyperparameters. (3) Representations of node pairs $\{v_i, v_j\}$ are combined into node-pair representations using the Hadamard product, i.e., $\mathbf{x}_{i,j} = \mathbf{x}_i \cdot \mathbf{x}_j$. (4) A binary Logistic Regression with 5-fold cross validation is trained using representations corresponding to edges and non-edges in $\hat{\mathbf{G}}$. (5) Classification performance is tested using representations of edges and non-edges of the original non-attacked graph $\mathbf{G}$. For computational efficiency, we approximate the performance using 5% of all possible node-pairs in $\mathbf{G}$. (6) We again repeat the complete process 3 times for non-deterministic attacks. For this task we report AUC and average precision scores.

**Experimental Setup.**    Our evaluation setup is structured as follows. First, in Section 4.3.1 we investigate the performance of node embedding approaches under random attacks. In this case, we use the *add_rand* and *del_rand* strategies and vary the attack budget $b \in [0.1, 0.2, ..., 0.9]$.[3] For node classification specifically, we report average results over $N_{tr} \in [0.1, 0.5, 0.9]$, 3 node shuffles for each $N_{tr}$ value, and 3 experiment repetitions for non-deterministic attacks. For network reconstruction we only perform the 3 experiment repetitions for non-deterministic attacks. We then also investigate the effect of the number of labeled nodes for node classification by comparing the results obtained for $N_{tr} = 0.1$ to $N_{tr} = 0.5$ and $N_{tr} = 0.9$. Second, in Section 4.3.2 we evaluate adversarial robustness. We use a similar setup with the following exceptions: we compare all attacks from Table 1 (random attacks are used as baselines) and the budget is fixed to $b = 0.2$. Third, in Section 4.3.3 we compare addition, deletion and rewiring attacks. For both downstream tasks we compare *add_rand*, *del_rand* and *rew_rand* and for node classification we additionally compare *add_ce*, *del_di* and *DICE*. Other parameters are set as for the adversarial attack experiment. In this section we also investigate differences between deletion attacks that disconnect and those that do not

---

[2]Exact hyperparameter values and method implementations are reported in the EvalNE configuration files.
[3]We acknowledge the impracticality of extreme budgets but find these edge cases theoretically interesting.

**Table 2:** Main statistics of the networks used for evaluation. The average degree is indicated by $\langle k \rangle$, the assortativity coefficient by $r$, and 'Viz' represents the network visualization task in Section 4.3.4.

| Network | Type | Task | # Nodes | # Edges | # Labels | $\langle k \rangle$ | $r$ |
|---------|------|------|---------|---------|----------|------|-----|
| Citeseer | Citation | NC | 2110 | 3668 | 6 | 3.48 | 0.01 |
| Cora | Citation | NC | 2485 | 5069 | 7 | 4.08 | -0.07 |
| PolBlogs | Web | NR | 1222 | 16714 | - | 27.35 | -0.22 |
| Facebook | Social | NR | 4039 | 88234 | - | 43.69 | 0.06 |
| IIP | Collaboration | Viz | 219 | 630 | 3 | 5.75 | -0.22 |
| StudentDB | Relational | Viz | 395 | 3423 | 7 | 17.33 | -0.34 |

disconnect the input networks. Lastly, in Section 4.3.4 we investigate the performance of common node classification baselines such as *DICE* that leverage full knowledge of the node labels.

## 4.2 Data

To conduct our experiments we use a total of 6 small and mid sized networks from different domains. Specifically, for node classification we use Citeseer [45] and Cora [46], two citation networks where nodes denote publications, edges represent citations between them and node labels indicate the main research field of each paper. For network reconstruction we use PolBlogs [47], a network of political blogs connected to each other via hyperlinks, and Facebook [48], a network of individuals and their social relations on the platform. Lastly, we perform qualitative and visualization experiments on the internet industry partnership (IIP) [49] and the StudentDB [50] networks. In the former, nodes represent companies, edges represent relations such as alliance or partnership and node labels indicate the company's main business area, i.e., user content, infrastructure or commerce. The latter, StudentDB, is a k-partite network representing a snapshot of the Antwerp University relational database. Nodes represent entities such as students, courses, tracks, etc., and edges are binary relations, e.g., student-in-track, course-in-track, etc. Node labels indicate the type of each node (see Appendix A.1 for more details). In Table 2 we summarize the main statistics of the networks used.

## 4.3 Experimental Results

### 4.3.1 Randomized attacks

We start in Figure 1a with node classification performance under random edge attacks and varying attack budgets. In the chart, negative budgets indicate edge deletion and positives indicate edge addition. In this case we allow edge deletions to disconnect the original networks. We report f1_micro scores for the Citeseer network (f1_macro results as well as those for the Cora network are similar and provided in Appendix A.2). From the figure we first note different general behaviors for edge deletion and addition attacks. Deletions cause a consistent performance degradation until complete network collapse at $b = 0.9$. Additions cause a sharper loss in performance for relatively low budget values ($b \leq 0.2$) which become less severe around $b = 0.4$. Thus, in the low budget regime commonly analyzed in the literature ($-0.2 < b < 0.2$), edge addition attacks are superior to edge deletion. Outside of this range, however, edge deletions are more damaging. This observation is reasonable given the asymmetry in the attack budgets. Removing 90% of the graph edges leaves significantly less information to learn an embedding from than adding 90% of spurious edges. We also observe from Figure 1a that NetMF and M-NMF are slightly more robust to edge additions than other approaches while edge deletion performance is similar across the board.

In Figure 1b we present the AUC scores for reconstructing the original Facebook network $\mathbf{G}$, from an attacked graph $\hat{\mathbf{G}}$. The plot indicates high edge recovery with AUCs $\approx 1$ despite the random attacks. Most methods maintain high robustness for a wide range of budget values. Some notable exceptions are Node2vec, LINE, and SDNE which consistently lose performance the more adversarial edges are added. One possible explanation is that these methods are not only affected by the addition of spurious edges but also by the removal of potentially informative negative samples, used by all three approaches to learn embeddings. For the PolBlogs dataset presented in Appendix A.2, we observe similar patterns. An exception in both networks is HOPE, which significantly degrades performance for strong edge deletion attacks ($b \leq -0.6$). This indicates the method is less suited to learning embeddings of highly sparse networks. The high robustness exhibited by the evaluated

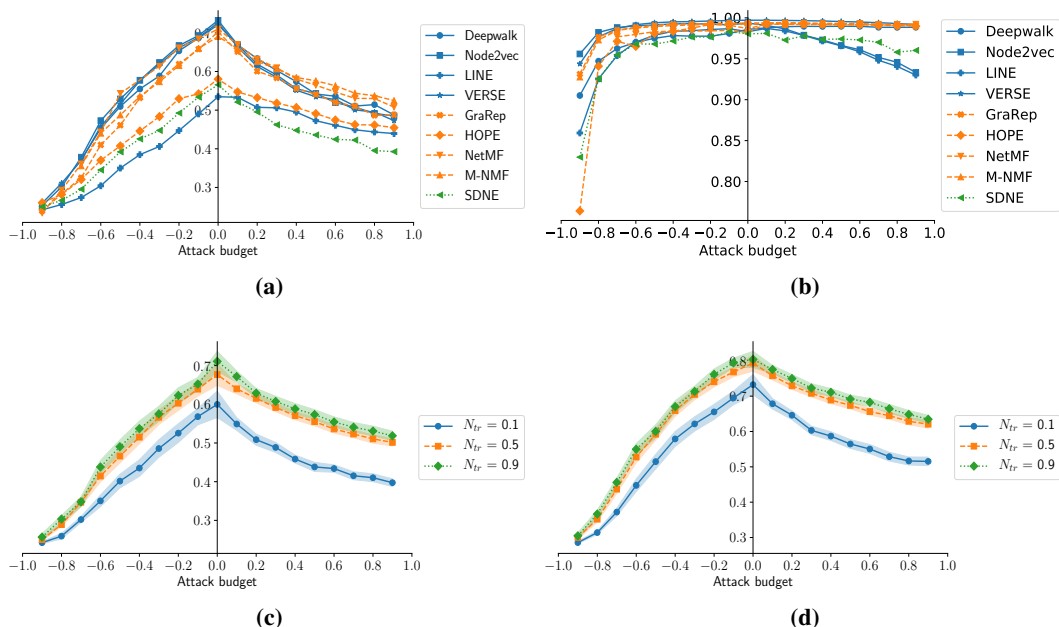

**Figure 1:** Robustness to randomized attacks for different budget values. The x-axis shows budgets as a fraction of all edges in the graph. Negative values represent edge deletion and positives edge addition attacks. Figure 1a presents f1_micro scores for node classification on Citeseer. Figure 1b shows AUCs for network reconstruction on Facebook. In Figures 1c and 1d we show average node classification f1_micro scores for different fractions of labeled nodes $N_{tr}$ on Citeseer and Cora, respectively. Shaded areas denote 95% confidence intervals.

approaches on this task is particularly interesting given the double impact of the attacks. Unlike in node classification, attacks on network reconstruction affect the models both at embedding learning time and binary classifier training (edge and non-edge train labels are obtained from the attacked $\hat{\mathbf{G}}$).

We now focus our attention on the impact of the number of train labels available for node classification ($N_{tr}$). In Figures 1c and 1d we compare the average performance over all methods and experiment repetitions for $N_{tr} \in [0.1, 0.5, 0.9]$ on Citeseer and Cora, respectively. For both networks we observe equally low performances when few labeled nodes are available i.e., $N_{tr} = 0.1$. For larger values ($N_{tr} \geq 0.5$) the performances are very similar. We also observe that as networks become denser (as we move right on the x-axis in each plot) the difference between low and high values of $N_{tr}$ become more significant. This indicates that node embedding methods will generally not provide robust predictions when few labeled nodes are available and this situation will worsen the denser the network is.

### 4.3.2 Adversarial attacks

We now compare the effect of different heuristic-based adversarial attacks on node classification. Figures 2a and 2b summarize the results on the Citeseer network for edge deletion and addition attacks, respectively. In both cases we present decreases in f1_micro caused by different attacks with budget $b = 0.2$, as compared to the performance on the non-attacked graph. Firstly, if we compare across graphs we observe that edge additions decrease performance more than deletions across all methods for this particular budget value. This is also consistent with our observations from Figure 1a for random attacks on node classification. Among the edge deletion attacks we see that *del_dd* is, from an adversarial perspective, the most effective strategy. With this attack, we are targeting edges from high degree to low degree nodes further increasing the uncertainty regarding the latter. On the other hand, for edge addition the most effective strategies are connecting edges with different labels together (*add_ce*) or connecting nodes with similar degrees to each other (*add_deg*). It is interesting to note that attacks with full knowledge of the node labels *del_di* and *add_ce* are not significantly

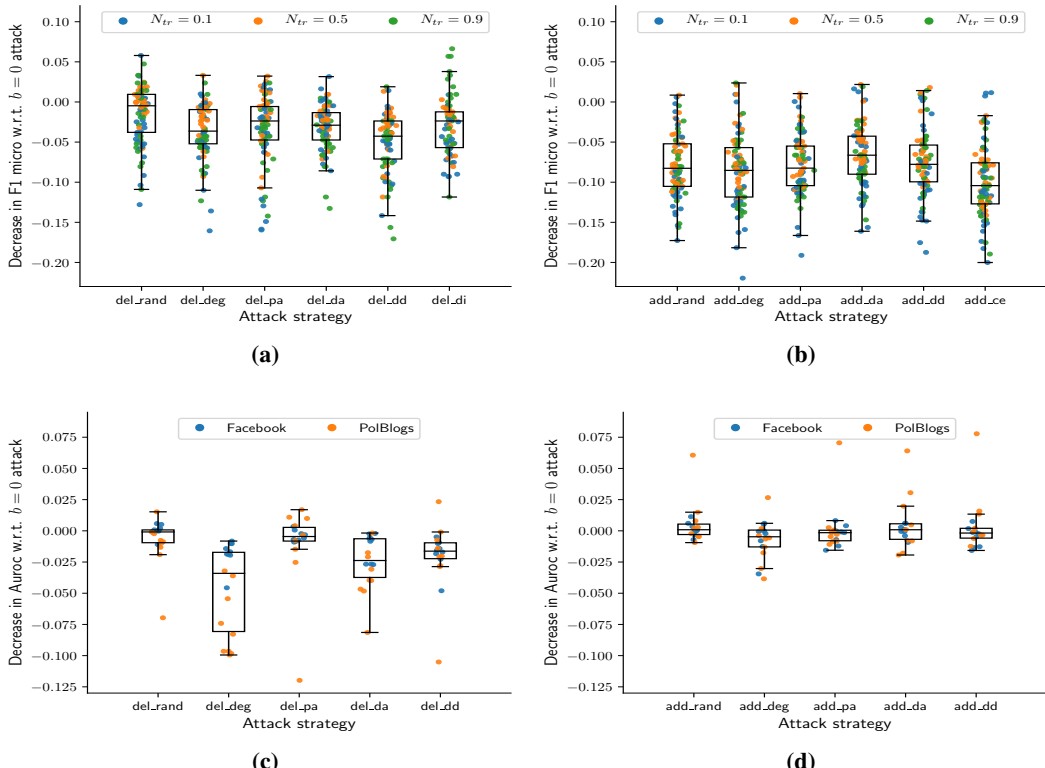

**Figure 2:** Comparison of adversarial edge deletion and addition attacks for $b = 0.2$. Figures 2a and 2b show deletion and addition attacks on node classification for Citeseer. Colors indicate the fraction of train nodes $N_{tr}$. Figures 2c and 2d show similar results for network reconstruction on both Facebook and PolBlogs networks combined (colors indicate the network).

stronger than others e.g., degree based attacks. The colors in both figures indicate different fractions of labeled nodes. We observe that most of the variance in performance comes from the experiments with $N_{tr} = 0.1$ (blue points) and that these are also mostly concentrated in the lower ends of the boxplots. The variances for $N_{tr} \geq 0.5$ are very similar across different attack strategies and networks.

In Figures 2c and 2d we present similar results for network reconstruction. In these cases we show the combined performances for both Facebook and PolBlogs datasets. The experiments reveal that edge deletion attacks are marginally stronger than edge addition. In particular, deleting edges based on degree is the most effective adversarial technique of the ones we have evaluated. Overall, we also observe much less variance in performance compared to the results on node classification.

### 4.3.3 Addition, deletion and rewiring attacks

In Figure 3 we compare edge addition, rewiring and deletion attacks on both downstream tasks. The attack budget is fixed to $b = 0.2$ and we show combined results for the two networks used in each task (marker color denotes the data used). We observe that for node classification rewiring attacks perform best (central boxes in the left and middle plots in Figure 3). This is also the case if we look at each individual dataset with results for Cora (orange dots) being significantly higher than those on Citeseer (blue dots). For network reconstruction we have much less data available, considering that we do not need to test different train sizes and shuffles per size. In this case the results indicate similar performances for all attack types. We further observe that results on the Facebook network are overall higher than on PolBlogs. The f1_macro and average precision scores for each task also corroborate this findings and are presented in Appendix A.3.

We further investigate how strong a role network connectivity plays in adversarial attacks. We compare random and degree attacks constrained to not disconnecting the input networks and their

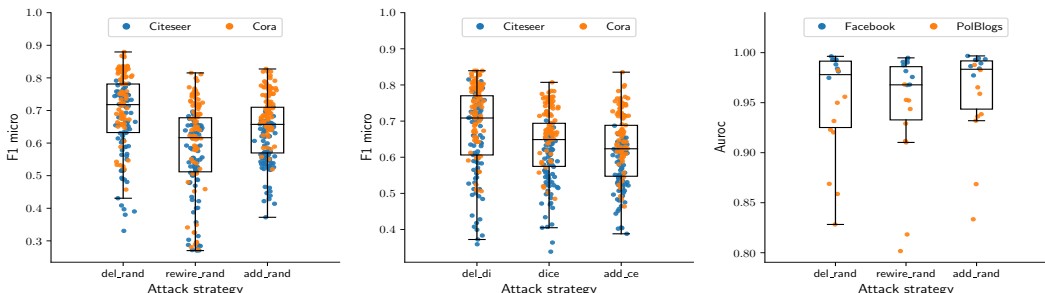

**Figure 3:** Comparison of edge addition, rewiring and deletion attacks for both downstream tasks. The leftmost and center figures present f1_micro scores for random and node label based attacks on node classification. The rightmost figure shows AUC results for random attacks on network reconstruction.

unconstrained counterparts. We find that constrained attacks are on average, over all methods and networks 5% less effective. Specifically, for random attacks the f1_micro performance without disconnections is $0.651 \pm 0.166$ (mean and standard deviation) and with disconnections $0.612 \pm 0.161$. Similarly, for degree based attacks average performance reaches $0.637 \pm 0.163$ when disconnections are prevented and $0.606 \pm 0.164$ when they are not.

### 4.3.4 Attacks exploiting node labels

In this section we investigate adversarial attacks on node classification where the attacker has full access to the node labels. These types of attacks e.g., DICE, are commonly used as baselines under the assumption that access to the node labels inevitably leads to stronger attacks. Here, we demonstrate that the above assumption does not always hold. Specifically, we find that node label homophily/heterophily has a strong impact on the performance of these types of attacks.

In this experiment we use the IIP and StudentDB datasets. The former is an example of a homophilic network where 70.9% of edges connect nodes of the same label. On the other hand, StudentDB is a strongly heterophilic network where no edges connect nodes sharing the same label. We summarize the results for node2vec, although our findings apply to other methods capturing high order proximities in graphs. We use *DICE* as an attack strategy.

We start our evaluation by attacking both networks with budgets $b \in [0.0, 0.2, 0.6]$. We then learn node embeddings and perform downstream node classification for each network and attack budget. Correctly and incorrectly classified nodes at validation time are recorded for each case. Figure 4 presents a spring-layout representations of the IIP network for each attack budget where nodes are colored based on their prediction status, correct (blue) or incorrect (orange). From the figure we can visually confirm that, as the attack strength increases, the misclassification rate (*mr* in the figure) also increases. This is also confirmed numerically by the *mr* value presented above each plot.

In Figure 5 we present the same information for the StudentDB network. In this case, as the attack strength increases the misclassification rate decreases (as can be seen visually and through the mr values). This seemingly counter intuitive behavior can be explained by the fact that *DICE* introduces additional information in the network reinforcing the heterophily schema (similar nodes remain unconnected while dissimilar ones are more connected). This dilutes the local network structure and makes nodes of the same type more similar to each other. Methods such as node2vec able to capture high order proximities between nodes can capitalize on this additional information to provide embeddings more suitable for node classification.

## 5   Conclusions

In this paper we have demonstrated that node embedding approaches, regardless of their underlying representation mechanisms, are sensitive to random and adversarial poison attacks. We have shown that results on downstream node classification are significantly less robust compared to those on network reconstruction. Our experiments also revealed that for low attacks budgets (below 20% of edges in the graph) edge addition attacks are generally stronger than edge deletions. Outside of this range, the opposite is true. Surprisingly, our empirical evaluation showed no significant differences

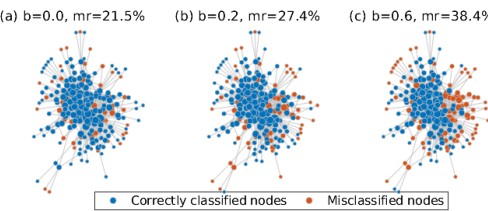
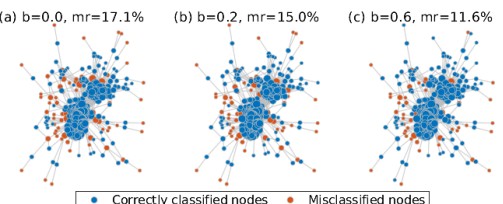

**Figure 4:** Correctly and incorrectly classified nodes for the homophilic IIP network for varying attack budgets.

**Figure 5:** Correctly and incorrectly classified nodes for the heterophilic StudentDB network for varying attack budgets.

between the heuristic-based adversarial attacks evaluated. Leveraging full knowledge of the node labels when attacking node classification does also not yield significantly stronger attacks. Finally, we have also shown that the number of labeled nodes plays a fundamental role in node classification robustness, that rewiring attacks are generally stronger than addition or deletion independently, and that attacks leveraging node label information can result in improved representations of heterophilic networks. With this work and our extension to robustness evaluation for the EvalNE software, we hope to lay the foundations for further research in this area.

## Acknowledgements

The research leading to these results has received funding from the European Research Council under the European Union's Seventh Framework Programme (FP7/2007-2013) (ERC Grant Agreement no. 615517), and under the European Union's Horizon 2020 research and innovation programme (ERC Grant Agreement no. 963924), from the Flemish Government under the "Onderzoeksprogramma Artificiële Intelligentie (AI) Vlaanderen" programme, and from the FWO (project no. G0F9816N, 3G042220).

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

# A  Appendix

## A.1  Further dataset details

The IIP network represents a set of companies competing in the internet industry between 1998 and 2001. Nodes in the graph denote companies and edges represent business relations such as joint venture, strategic alliance or other type of partnership. The associated node labels denote the company's main business area i.e., content, infrastructure of commerce.

The StudentDB network represents a snapshot of Antwerp University's relational student database. Nodes in the network represent entities, more specifically: students, professors, tracks, programs, courses and rooms. Edges constitute binary relations between them, that is, student-in-track, student-in-program, student-takes-course, professor-teaches-course, and course-in-room. Numerical node labels are assigned according to each node's type.

## A.2  Randomized attacks: additional results

In this section we present our additional experiments regarding randomized attacks on node embeddings. We start in Figures 6 and 7 by presenting the node classification f1_micro results for the Cora dataset and the network reconstruction AUC scores for PolBlogs.

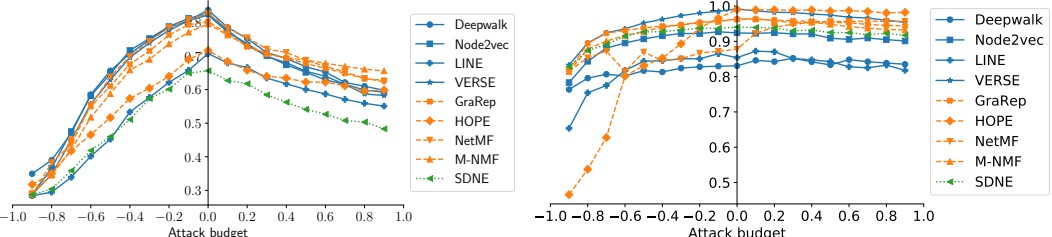

**Figure 6:** Node classification performance for the Cora network. Y axis indicates f1_micro scores. Negative attack budgets indicate edge deletion.

**Figure 7:** Network reconstruction performance for the PolBlogs network. Y axis indicates AUC scores. Negative attack budgets indicate edge deletion.

In Figures 8 and 9 we summarize the f1_macro scores for both Citeseer and Cora and Figures 10 and 11 present the average precision on Facebook and PolBlogs.

## A.3  Other attacks: additional results

We also compare the performance of edge addition, rewiring and deletion on both downstream tasks in terms of f1_micro and average precision. These results support our conclusions in Section 4.3.3 (see Figure 12).

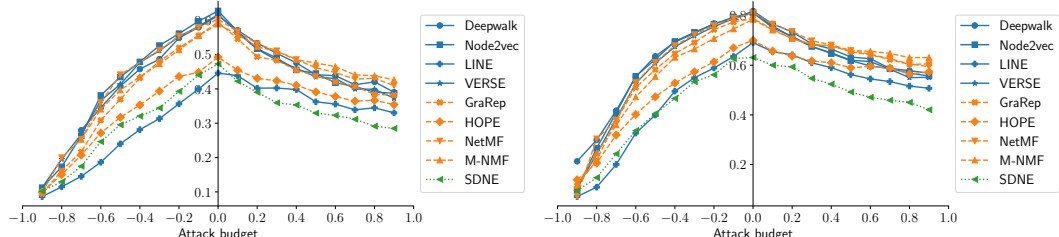

**Figure 8:** Node classification performance for the Citeseer network. Y axis indicates f1_macro scores. Negative attack budgets indicate edge deletion.

**Figure 9:** Node classification performance for the Cora network. Y axis indicates f1_macro scores. Negative attack budgets indicate edge deletion.

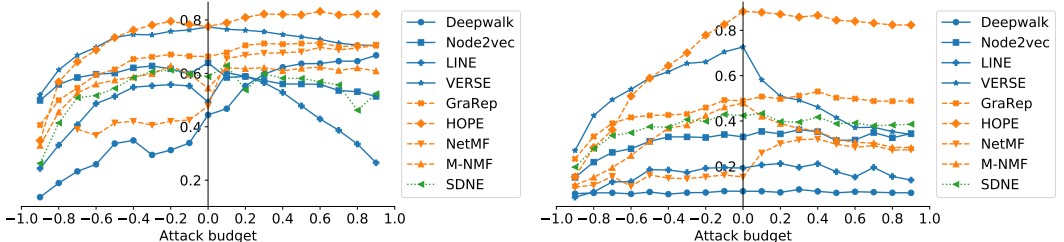

**Figure 10:** Network reconstruction performance for the Facebook network. Y axis indicates average precision scores. Negative attack budgets indicate edge deletion.

**Figure 11:** Network reconstruction performance for the PolBlogs network. Y axis indicates average precision scores. Negative attack budgets indicate edge deletion.

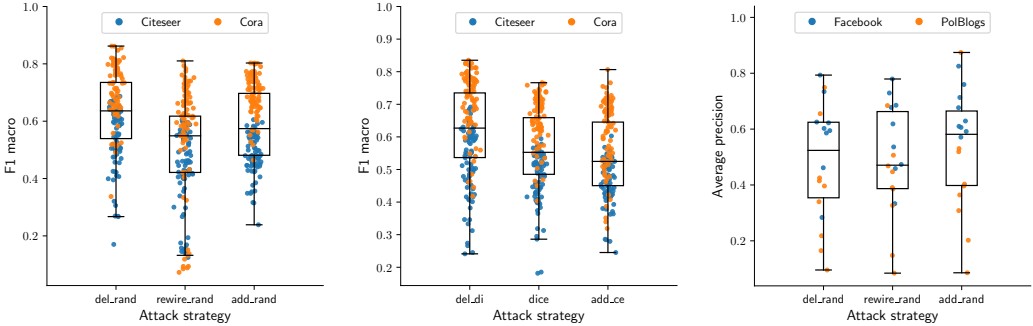

**Figure 12:** Comparison of edge addition, rewiring and deletion attacks for both downstream tasks. The leftmost and center figures present f1_macro scores for random and node label based attacks on node classification. The rightmost figure shows average precision results for random attacks on network reconstruction.

