# OpenReview forum: "A Systematic Evaluation of Node Embedding Robustness"
_logconference.io/LOG/2022/Conference — LoG 2022 Poster_

### Official Review · Reviewer_L8BN · 2022-10-12

**Overall Score:** 5
**Confidence:** 4

**Review:**

**Paper Summary**

This work aims to empirically evaluate the robustness of node embedding models under graph attacks. Specifically, authors first apply different graph attacks to the input graph by inserting/deleting/rewiring edges. Then, they evaluate the performance of 10 unsupervised embedding models on node classification and network reconstruction tasks. Based on the experiments, authors draw some conclusions such as degree-based and label-based attacks are the most effective ones for attacking the node embedding models.

**Strong Points**

  - Overall, the paper is well-written and easy to follow.
  - Authors have provided detailed experimental setup and configuration files for reproducing the results.


**Weak Points**

  - Authors only evaluate the model robustness against some weak and heuristic attacks (e.g., DICE). However, DICE has been shown to perform much worse than other attacks (e.g., PR-BCD and GR-BCD) [1]. Without evaluating model robustness against strongest attacks, the contribution of this work is limited.
  - This work only focuses on undefended node embedding models, which are already known to be vulnerable to graph attacks [2]. It is more important and interesting to systematically evaluate the robustness of defended embedding methods (e.g., the graph purification method proposed in [3]).
  - Some of authors' claims seem to be not fully convincing or even contradict to the experimental results. Please see the questions below for details.


[1]: Geisler et al. “Robustness of Graph Neural Networks at Scale.” NeurIPS'21. \
[2]: Bojchevski et al. “Adversarial Attacks on Node Embeddings via Graph Poisoning.” ICML'19. \
[3]: Entezari et al. “All You Need Is Low (Rank): Defending Against Adversarial Attacks on Graphs.” WSDM'20.

**Questions**

  - Authors claim that Skip-Gram methods are more robust to edge addition than other approaches. However, Figure 1.a shows that the matrix factorization methods (NetMF and M-NMF) outperform Skip-Gram methods against edge addition. Could authors explain this?
  - From Figure 1.b, authors observe that Skip-Gram methods consistently perform worse than other models for the network reconstruction task, when more edges are added to the network. Is there any insight about this observation?
  - Authors conclude that the results in Figure 2.a are very similar for $N_{tr} \ge 0.5$. However, it is clearly shown in Figure 2.a that green points ($N_{tr} = 0.9$) are spread out in the boxplots, which means they have very dissimilar results per attack method. How do the authors draw the conclusion that the results are similar?
  - The explanation of results on heterophilic graphs (lines 326~328) seems not compelling. Why edge additions result in a more compact representation? What is the exact definition of the compact representation here? Why the compact representation leads to lower margins for the classifier?
  - Based on the observations authors made in this paper, could authors come up with a concrete strategy for improving either existing attack or defense methods?


**Typos**

  - Line 160: asses => assess.
  - Figure 2: "b=0 attack" => "b=0.2 attack", "Figures 2a and 2b" => "Figures 2c and 2d".
  - Figures 2.c and 2.d are uncolored.

**Recommendation**

Although it is interesting to evaluate node embedding robustness, this work ignores strong attacks and defended embedding models during robustness evaluation. Besides, some authors' claims based on empirical results are not fully convincing. Thus, I tend to reject this paper.

---

### Official Review · Reviewer_RYNg · 2022-10-13

**Overall Score:** 6
**Confidence:** 3

**Review:**

## Overview:
The authors perform a thorough empirical evaluation of the robustness of a number of current unsupervised node embedding methods to heuristic poisoning attacks. They consider two downstream attacks to test the effectiveness of the considered poisoning attacks -- node classification and network reconstruction. The authors provide an in-depth qualitative and quantitative analysis of the inherent robustness to poisoning of a number of node embedding methods.

## Strengths:
While the paper does not introduce "novel" ideas in the field of poisoning attacks on graphs, I believe their detailed comparison of existing methods is valuable to the community, akin to [1] for poisoning/backdoor attacks on computer vision systems.

Their experimental setup is simple and general enough to allow for a wide variety of node embedding methods.

## Weaknesses:
It seems unnatural to me that the authors have chosen to not consider any optimization-based attacks. The authors cite the superior computational efficiency of the heuristic attacks as a motivation for that choice. However, it is not clear to me whether optimization-based attacks can be significantly more powerful/effective than the heuristic attacks considered in this paper. In particular, I believe that a stronger argument is needed to motivate the choice of not considering optimization-based attacks in the evaluation.

## Comments/questions:
- The authors report results on homophilic and heterophilic graphs in Section 4.3.4. Why is it important to study the effect of label homophily and heterophily on robustness?
- Practicality of attack budget: Certain phenomena (e.g. structural collapse) discussed in the paper appear only at extremely high levels of poisoning (e.g. 90% of edges being deleted). This seems outside the range of "reasonable" budget for an adversary within the poisoning literature, e.g. [2].
- Why is the budget for adversarial attacks fixed at $b=0.2$ (we allow a $0.2$ of the edges to be added/deleted/re-wired), while a sweep of attack budgets is considered for randomized attacks? I believe a figure in the style of Figure 1 for adversarial attacks would provide a much more transparent comparison between randomized and adversarial attacks.

## Minor comments:
- Line 53: that -> than

## Recommendation:
I believe that this paper provides a useful step towards a unified benchmark for poisoning attacks in the context of graph learning (and node embeddings in particular). Thus despite the stated weaknesses, I recommend acceptance. I am willing to reconsider my score if the stated questions/comments are addressed.

[1] Schwarzschild, Avi, et al. "Just how toxic is data poisoning? a unified benchmark for backdoor and data poisoning attacks." International Conference on Machine Learning. PMLR, 2021.

[2] Biggio, Battista, Blaine Nelson, and Pavel Laskov. "Poisoning attacks against support vector machines." arXiv preprint arXiv:1206.6389 (2012).

---
# Post-rebuttal

I am overall satisfied with the proposed changes (and curious to see the equivalent of Figure 1 for adversarial attacks). Thus I keep my recommendation of acceptance.

---

### Official Review · Reviewer_HC1T · 2022-10-18

**Overall Score:** 6
**Confidence:** 3

**Review:**

This paper focuses on node embedding robustness, a very interesting topic. The authors perform a systematic evaluation of node embedding robustness with 10 node embedding approaches and 14 different poison attack strategies on 6 datasets. Their findings reveal that, compared to network reconstruction tasks, node classification tasks are more vulnerable to poison attacks.

I have the following comments to further improve the paper.

1. The authors do not consider optimization-based attacks due to the higher computational cost. However, to give a more comprehensive view of the embedding robustness, I would suggest the authors evaluate optimization-based attacks as well since those attacks are usually optimized with global information and thus can achieve better performance.

2. The authors provide a macro-level embedding robustness evaluation. However, it's unclear what kind of nodes are more vulnerable to attacks. Therefore I would suggest the authors conduct a deeper analysis of the micro-level embedding robustness as well (e.g., separate the nodes into different groups by the network properties like degree, assortativity, and labels).

3. I would appreciate it if the authors could provide more insight regarding why edge deletion is more effective than edge addition.

---

### Official Review · Reviewer_xmL1 · 2022-10-22

**Overall Score:** 6
**Confidence:** 4

**Review:**

This paper provides a systematic evaluation of some random and adversarial attack heuristics to study the robustness of unsupervised node embedding methods. While multiple attack heuristics are compared, the paper should be further improved by extending its scope to discuss more advanced graph poisoning attack methods in the unsupervised setting. Meanwhile, it would be better if the authors can provide more fundamental and instructive insights from the empirical results. My detailed comments are as follows.

1. The adversarial attack strategies in Table 1 should be defined more formally and mathematically, otherwise it is still unclear for the readers to understand how these attacks perform exactly, judging from the natural language description. For example, the authors should explicitly provide the definition of assortativity so the readers can understand it without referring to the other paper, and the author should explain how add_da and add_dd are performed exactly to increase and decrease this property. Meanwhile, the author should provide proper intuition or justification about why each of these attacks can be effective.

2. The set of adversarial attacks studied in this paper are simple heuristics, while several unsupervised adversarial attack methods are recently studied, which should also be discussed. For example, the structure is perturbed to maximally influence the low-rank optimization of matrix factorization based methods in [1, 2]; the edges are flipped to maximally change the spectral property of a graph in [3, 4]. I cannot see the reason why optimization-based attacks are beyond the scope of this paper, as this paper claims to be a systematic evaluation of unsupervised node embedding robustness. The optimization-based methods can better locate the most important edges that maximally change the structural property of the graph or the outcome of node embedding methods. The necessity of considering these state-of-the-art attacks can be also reflected by the results: the studied simple adversarial attack heuristics are not effective compared with random ones.

3. What is the meaning of “Viz” in Table 2? It is used without definition.

4. The conclusion in line 246 that skip-Gram methods are more robust seems not true, as matrix factorization methods have similar curves as skip-gram methods, judging from Figure 1(a).

5. What are the most important takeaways from the evaluations that can help the community? The authors should better highlight such insights.

6. The finding that adding more edges for heterophilic graphs decreases the misclassification rate is interesting. However, the conjecture of introducing new edges can lower the margin of the classifier should be verified empirically or theoretically.

[1] Sun, Mingjie, et al. "Data poisoning attack against unsupervised node embedding methods." arXiv preprint arXiv:1810.12881 (2018).

[2] Bojchevski, Aleksandar, and Stephan Günnemann. "Adversarial attacks on node embeddings via graph poisoning." International Conference on Machine Learning. PMLR, 2019.

[3] Lin, Lu, Ethan Blaser, and Hongning Wang. "Graph Structural Attack by Perturbing Spectral Distance." Proceedings of the 28th ACM SIGKDD Conference on Knowledge Discovery and Data Mining. 2022.

[4] Chang, Heng, et al. "A restricted black-box adversarial framework towards attacking graph embedding models." Proceedings of the AAAI Conference on Artificial Intelligence. Vol. 34. No. 04. 2020.

**Post-rebuttal comments**
The authors addressed most of my concerns, and I have raised my score.

---

### Meta-Review · Area_Chair_cuzC · 2022-11-17

**Confidence:** 4
**Recommendation:** Accept

**Meta Review:**

This is an empirical paper investigating the robustness of node embedding methods against topological perturbations. Reviewers generally agreed a thorough empirical investigation like presented in the paper is valuable for the graph learning community. On the other hand, the analysis can probably be further strengthened by considering more advanced (e.g., optimisation-based) attacks or defence models. It would also be good to dive deeper into the structural patterns of the attacks to gain more insight into robustness. On this point, I suggest the authors discuss further related work in the literature, e.g., [1] (Section 7.2) and [2] (Section 5) which both touch upon the issue of structural patterns of attacks.

Although there are certain aspects for improvement, my overall assessment is that the positives of the paper outweigh the negatives. Personally I would also like to encourage more work in the space of understanding robustness of graph learning models from a structural perspective. In this sense I believe the paper is, despite its simplified setting, a valuable contribution to the community. Therefore I am inclined to accept the paper.

References:
[1] https://arxiv.org/pdf/2003.00653.pdf
[2] https://arxiv.org/pdf/2111.02842.pdf

---

### Decision · Program_Chairs · 2022-11-22

**Decision:**

Accept (Poster)

**Comment:**

The PCs agree with the AC assessment. We encourage the authors to improve their paper in the camera-ready version by incorporating additional comments raised during the review phase.